# Differentiable Distance Between Hierarchically-Structured Data

## Abstract

Many machine learning algorithms solving various problems are available for metric spaces. While there are plenty of distances for vector spaces, much less exists for structured data (rooted heterogeneous trees) stored in popular formats like JSON, XML, ProtoBuffer, MessagePack, etc. This paper introduces the Hierarchically-structured Tree Distance (HTD) designed especially for these data. The HTD distance is modular with differentiable parameters weighting the importance of different sub-spaces. This allows the distance to be tailored to a given dataset and task, such as classification, clustering, and anomaly detection. The extensive experimental comparison shows that distance-based algorithms with the proposed HTD distance are competitive to state-of-the-art methods based on neural networks with orders of magnitude more parameters. Furthermore, we show that HTD is more suited to analyze heterogeneous Graph Neural Networks than Tree Mover's Distance.

## 1 Introduction

Most machine learning tasks can be approached by algorithms relying on the existence of distance. These tasks include classification (Fix & Hodges, 1951), anomaly detection (Breunig et al., 2000; Knorr et al., 2000), clustering (Rdusseeun & Kaufman, 1987; Sibson, 1973)), dimensionality reduction for visualization (McInnes et al., 2018), indexing methods for fast retrieval (Zezula et al., 2006), explanation (Chen et al., 2019; Guidotti, 2022), and density estimation (Williams & Rasmussen, 2006). A suitable distance on a dataset of interest therefore makes all this vast prior art readily available for solving downstream tasks. Distance is also essential for studying theoretical properties of algorithms (Chuang & Jegelka, 2022).

While for Euclidean spaces distances are well known, it is much harder to define them on objects with variable dimensional objects such as trees or graphs. A particularly important but neglected type of objects are those stored in structured data formats such as JSON, XML, or Protobuffer. These formats are popular among engineers since they allow them to logically organize data with increasing levels of detail, which is natural for humans. Moreover, the contemporary internet experience relies on exchanging messages stored in these data formats.

Data stored in structured data formats, further called HS-Trees are rooted trees of fixed depth, where a large number of nodes have different semantics and structure and where some nodes can have a fixed number of edges (and child). These properties were exploited in some supervised learning methods (Socher et al., 2011; Shuai et al., 2016; Tai et al., 2015; Cheng et al., 2018; Woof & Chen, 2020) offering properties not available for general graphs, such as theoretical guarantees due to an extension of approximation theorem (Pevny & Kovarik, 2019), and low computational complexity, as single pass from leaves to root (Mandlík et al., 2022) is sufficient. Furthermore as shown in Chuang & Jegelka (2022) for graph neural networks (GNN) based on message passing process samples from HS-Trees when the computation graph is unrolled.

Despite the practical importance and ubiquity of HS-Trees, there is very little prior art about distance on HS-Trees. In Šopík & Strenáčik (2022) (further called TED) tree-edit

Table 1: Properties of distance functions on attributed trees.

| | Differentiable | Heterogeneous | Metric | Free Parameters | Modular |
|---|:---:|:---:|:---:|:---:|:---:|
| Tree Mover's Distance (TMD) | ✓ | ✗ | ✗ | ✓ | ✗ |
| Tree Edit Distance (TED) | ✗ | ✓ | ✓ | ✗ | ✗ |
| Hierarchically-structured Tree D. (HTD) | ✓ | ✓ | ✓ | ✓ | ✓ |

distance is extended to HS-Trees. TED distance is parametrized by costs, but they are non-differentiable, which complicates their optimization (metric learning) by efficient first-order methods. Tree Mover's Distance (TMD) (Chuang & Jegelka, 2022), proposed for rooted homogeneous trees to study the generalization properties of GNNs, does not support heterogeneous data. To address these shortcomings, this paper proposes HTD distance, which exploits the recursive nature of the data format, allowing **modular** construction by combining potentially different metrics on different levels of the tree. HTD distance is **parametrized by weights**, which control importance on different parts. The distance is **differentiable**, so it can be seamlessly incorporated into many modern algorithms, especially in those optimizing the metric for the given problem (metric learning). The computation complexity depends on the construction, specifically on the used distance on multisets. The most general setting with Wasserstein distance has cubic complexity, but for many practical problems, it is sufficient to use Haussdorf distance or Chamfer pseudo-distance with quadratic complexity.

The performance of HTD distance is experimentally evaluated on i) supervised learning, ii) anomaly detection, iii) analysis of heterogenous GNNs, (iv) clustering (presented in the appendix due to lack of space), and (v) inside UMAP for visualization. The experimental results show that distance-based algorithms with the proposed distance are competitive (and frequently better) to state-of-the-art methods based on neural networks (Pevny & Kovarik, 2019; Mandlík et al., 2022) while having a few orders of magnitude fewer parameters. We also show that the HTD better correlates with the performance of GNNs for heterogeneous graphs than Tree Mover's Distance with the homogenization (Chuang & Jegelka, 2022).

The paper is organized as follows. The next section formally defines HS-Trees and their relation to GNNs for heterogeneous graphs. Section 3 defines the HTD distance and discusses the impact of choices on its generality (theoretical guarantees) and computation complexity. Section 4 reviews the related work. Experimental comparison on classification, anomaly detection, analysis of GNNs, and application to visualization is shown in Section 5. The last section summarizes the paper.

## 2 BACKGROUND

This section first defines *schema*, which is an important concept in the definition of HS-Trees, and then shows their relation to the computation graph of GNNs. The relation of HS-Trees to data stored in structured formats, like JSON, is left to the Appendix C.

The HTD distance is defined for samples with the same *schema*. Schema corresponds to "data type" in programming languages, message type in protocol buffers, schema in JSON (Pezoa et al., 2016), and document type definition in XML files (Farrell & Lausen, 2007). Schema defines the *set of possible values*, their *semantics*, and the structure of the data (type of nodes and their branching). To prevent confusion, schemas are always denoted by blackboard letters. $x \in \mathbb{S}$ denotes that sample $x$ is from the schema $\mathbb{S}$, but one may also say that sample $x$ has schema $\mathbb{S}$.

The definition requires the introduction of *elementary data types*, which are simple data types like numbers, tensors of fixed dimension, categorical variables, and strings. A second key part of the schema is multiset, denoted as $[\![\cdot]\!]$, which corresponds in structured formats to unordered arrays with possibly repeated elements. The third key component is the dictionary (hashmap), which introduces heterogeneity into the data. Formal definition follows.

**Definition 2.1** (Schema). *The set of all schemas $\mathcal{S}$, and the element of relation '$\in$' is defined recursively as follows:*

    *1.* **Leaves***: Let $\mathbb{L}$ be an elementary data type. Then $\mathbb{L} \in \mathcal{S}$.*

*It holds $x \in \mathbb{L}$ if and only if $x$ is of data type $\mathbb{L}$.*

2. **Bags**: *Let $\mathbb{A} = [\![\mathbb{S}]\!]$ where $\mathbb{S} \in \mathcal{S}$. Then $\mathbb{A} \in \mathcal{S}$.*

   *$x \in \mathbb{A}$ if and only if $x = [\![x_1, \ldots, x_n]\!]$, where $n \geq 0$ and $x_i \in \mathbb{S}$ for each $i = 1, \ldots, n$.*

3. **Dicts**: *Let $\mathbb{D} = \{(k_i, \mathbb{S}_i)\}_{i=1}^m$, where $K = \{k_i\}_{i=1}^n$ is a set of unique keys and $\mathbb{S}_i \in \mathcal{S}$ for all $i = 1, \ldots, m$. Then $\mathbb{D} \in \mathcal{S}$.*

   *$x \in \mathbb{D}$ if and only if $x = \{(k_i, x_i)\}_{i=1}^n$, where $(k_i, s_i) \in \mathbb{D}$ and $x_i \in \mathbb{S}_i$ for each $i = 1, \ldots, l$.*

HS-Trees is a union of all samples from all schema.

In Definition 2.1, Bags are used to represent multisets and sequences of arbitrary (including empty) size. They are assumed to be permutation invariant; therefore, the position has to be encoded through position encoding. Importantly, all items of the Bag have the same schema. Models accepting Bags need to handle inputs of arbitrary lengths (or size) requiring some form of aggregation which is either explicit through functions like mean, sum, and max (Muandet et al., 2012; Zaheer et al., 2017; Pevný & Somol, 2017) or through recurrence (Hochreiter & Schmidhuber, 1997). Dict represent Cartesian products of a fixed number of subspaces with a fixed schema. Neural networks processing Dict typically projects individual subspaces to a vector space and then *concatenate* the representations. The concatenation is impossible for Bags because they can have arbitrary sizes. Universal approximation theorem for HS-Trees has been proved in Pevny & Kovarik (2019).

## 2.1 Relation of HS-Trees to GNNs

The rest of this section emphasizes how the above definition of HS-Trees relates to computation graphs of GNNs based on message passing (Xu et al., 2019). Let $G = (\mathcal{V}, \mathcal{E})$ be a homogeneous graph with vertices with feature vectors $\{h_{v_i}^0\}_{i=1}^{|\mathcal{V}|}$, $h_{v_i}^0 \in \mathbb{R}^d$. GNNs update the representation of graph's $i^{\text{th}}$ vertex, $h_{v_i}^k$, in each ($k^{\text{th}}$) iteration according to the formula:

$$h_{v_i}^k = f^k \left( h_{v_i}^{k-1}, \operatorname{agg} \left( [\![ g^k(h_{v_j}^{k-1}) | v_j \in \mathcal{N}(v_i) ]\!] \right) \right), \tag{1}$$

where $f^k$ and $g^k$ are feed-forward neural networks, agg is an aggregation function (e.g. mean, max, sum), and $\mathcal{N}(v_i)$ denote the set of neighbors of $v_i$. The input to the update function (1) is *always* an ordered pair consisting of $h_{v_i}^{k-1}$ and the neighborhood $[\![ h_{v_j}^{k-1}) | v_j \in \mathcal{N}(v_i) ]\!]$, which corresponds to a Dict. The reason for using Dict instead of Bag with two items is that both children are *semantically and structurally different*. One represents the feature vector of the vertex, while the other that of all its neighbors'. They also have a different schema: if $h_i^{k-1} \in \mathbb{H}^{k-1}$ than the neighborhood $[\![ h_{v_j}^{k-1} | v_j \in \mathcal{N}(v_i) ]\!] \in [\![ \mathbb{H}^{k-1} ]\!]$). On the contrary, the neighborhood corresponds to the Bag because its size differs between vertices while its items are *semantically and structurally the same*, and they share the same schema. The sample updating the $h_{v_i}^k$ expressed as HS-Tree is therefore

$$\left\{ \operatorname{self} = h_{v_i}^{k-1}, \operatorname{neighborhood} = [\![ h_{v_j}^{k-1} | v_j \in \mathcal{N}(v_i) ]\!] \right\}. \tag{2}$$

For example in Fig. 1a, inputs to function updating vertices $v_0, v_1, \ldots, v_4$ in the first iteration are respectively

$$\left\{ \operatorname{self} = h_{v_0}^0, \operatorname{neighborhood} = [\![ h_{v_1}^0, h_{v_2}^0, h_{v_3}^0 ]\!] \right\}, \quad \left\{ \operatorname{self} = h_{v_1}^0, \operatorname{neighborhood} = [\![ h_{v_0}^0, h_{v_2}^0 ]\!] \right\},$$

$$\left\{ \operatorname{self} = h_{v_2}^0, \operatorname{neighborhood} = [\![ h_{v_0}^0, h_{v_1}^0 ]\!] \right\}, \quad \left\{ \operatorname{self} = h_{v_3}^0, \operatorname{neighborhood} = [\![ h_{v_0}^0 ]\!] \right\},$$

$$\left\{ \operatorname{self} = h_{v_4}^0, \operatorname{neighborhood} = [\![ \, ]\!] \right\}.$$

Inputs in subsequent iterations are obtained accordingly. Due to the recursive nature, they all belong to HS-Trees.

Let's now assume heterogeneous graph $G = (\{\mathcal{V}_r\}_1^l, \{\mathcal{E}_{rs}\}_{1,1}^{l,l})$, where $\mathcal{V}_r$ denotes the set of vertices of $r^{\text{th}}$ type and $\mathcal{E}_{rs}$ denotes set of edges between vertices $\mathcal{V}_r$ and $\mathcal{V}_s$. Extensions of GNNs to heterogeneous graphs (Schlichtkrull et al., 2018; Guan et al., 2024) update vertices of each type $\mathcal{V}_r$ based on neighborhoods in all types of vertices defined by sets of edges

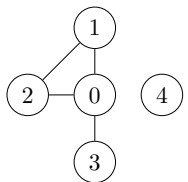 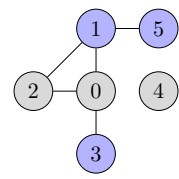

(a) Homogeneous graph

(b) Heterogeneous graph with two types of nodes distinguished by colors.

Figure 1: Illustrative examples of homogeneous and heterogeneous graph.

$\{\mathcal{E}_{rs}\}_{s=1}^l$. The update of a vertex $v_i$ of $r^{\text{th}}$ type in $k^{\text{th}}$ iteration can be written as

$$h_{v_i}^k = f_r^k \left( h_{v_i}^{k-1}, \text{agg}\left( [\![ g_{r1}^k(h_{v_j}^{k-1}) | v_j \in \mathcal{N}_{r1}(v_i) ]\!] \right), \dots, \text{agg}\left( [\![ g_{rl}^k(h_{v_j}^{k-1}) | v_j \in \mathcal{N}_{rl}(v_i) ]\!] \right) \right), \quad (3)$$

where $\mathcal{N}_{rs}(v_i)$ is a neighborhood of vertex $v_i$ of the $r^{\text{th}}$ type with vertices of type $s$ defined by edges $\mathcal{E}_{rs}$, and $f_r^k, g_{r1}, \dots, g_{rl}$ are feed-forward neural networks.

The input to the function updating the vertex of the $r^{\text{th}}$ type is, therefore, an ordered tuple (in HS-Trees represented as Dict) consisting of the representation of the vertex from the previous iteration and $l$ representations of the neighborhood with different types of vertices. Representing the input as a Dict puts each neighborhood in a different space, as they are semantically different. The computation graph again unfolds into a tree through recursion.

For example in Fig. 1b, samples updating grey vertices $\{v_0, v_2, v_4\}$ in the first iteration are

$$\left\{ \text{self} = h_{v_0}^0, \text{blue neigh.} = [\![ h_{v_1}^0, h_{v_3}^0 ]\!], \text{gray neigh.} = [\![ h_{v_2}^0 ]\!] \right\},$$
$$\left\{ \text{self} = h_{v_2}^0, \text{blue neigh.} = [\![ h_{v_1}^0 ]\!], \text{gray neigh.} = [\![ h_{v_0}^0 ]\!] \right\},$$
$$\left\{ \text{self} = h_{v_4}^0, \text{blue neigh.} = [\![\,]\!], \text{gray neigh.} = [\![\,]\!] \right\},$$

and those updating blue vertices $\{v_1, v_3, v_5\}$ are

$$\left\{ \text{self} = h_{v_1}^0, \text{blue neigh.} = [\![ h_{v_5}^0 ]\!], \text{gray neigh.} = [\![ h_{v_0}^0, h_{v_2}^0 ]\!] \right\},$$
$$\left\{ \text{self} = h_{v_3}^0, \text{blue neigh.} = [\![\,]\!], \text{gray neigh.} = [\![ h_{v_0}^0 ]\!] \right\},$$
$$\left\{ \text{self} = h_{v_5}^0, \text{blue neigh.} = [\![ h_{v_1}^0 ]\!], \text{gray neigh.} = [\![\,]\!] \right\}.$$

The input to the readout function is a Dict with $l$ items, each being a Bag containing representations of vertices of a given type.

## 3 METRIC ON HS-TREES

This section presents the HTD on the space of HS-Trees with the same schema. The construction is recursive and assumes the existence of distances on leaves, bags, and dictionaries. These choices determine the final properties. Therefore, they are discussed first, and then HTD is defined in Section 3.4.

### 3.1 METRIC ON LEAVES

Leaves contain various elementary data types (real numbers, tensors of fixed dimensions, categorical variables, and strings). It is assumed the distance on Leaves exists, and its definition is outside of the scope of this work, but the most common ones are listed below.

Distances between tensors on Euclidean spaces are usually measured by distances induced by $L_p$ norms, of which $L_1$ and $L_2$ are the most popular. Categorical data are usually encoded as one-hot vectors, and distances induced by $L_p$ norms collapse to zero / one if two values are equal/unequal. Strings offer two conceptually different approaches. The first, such as the Levenshtein (Levenshtein, 1966) or Jaro-Winkler distance (Winkler, 1990), are defined directly on the space of all strings. A popular alternative is to measure the distance in the

Table 2: Short overview of various Bag distances, their acronyms, computational complexity, and whether they are proper metrics.

| name | acronym | complexity | metric |
|------|---------|------------|--------|
| Wasserstein Distance | $d_{\text{WA}}$ | $\mathcal{O}(n^3)$ | yes |
| Partial Wasserstein D. | $d_{\text{PW}}$ | $\mathcal{O}(n^3)$ | yes |
| Hausdorff Distance | $d_{\text{HA}}$ | $\mathcal{O}(n^2)$ | yes |
| Chamfer Distance | $d_{\text{CH}}$ | $\mathcal{O}(n^2)$ | no |

Euclidean space to which the strings are projected, for example, by word2vec (Mikolov et al., 2013), BERT (Devlin et al., 2019), or N-Grams (Hiemstra, 2009). While the latter approach may not be proper distance on the space of strings, it better captures semantic similarity.

## 3.2 METRIC ON DICTIONARIES

Dicts can be viewed as a Cartesian product, which makes the product metric (Deza & Deza, 2009) a natural choice. To calibrate ranges of distances on different sub-spaces (corresponding to different keys in the dictionary), we introduce weights $w_i$, which are also used to reflect the importance of individual parts. The resulting *Weighted Product Metric* ($d_{\text{WPM}}$) is defined as

**Definition 3.1** (Weighted product metric ($d_{\text{WPM}}$)). *Let* $\{(\mathbb{M}_i, d_i)\}_{i=1}^n$ *be a set of arbitrary metric spaces, then* $d_{\text{WPM}} : (\mathbb{M}_1, \ldots, \mathbb{M}_n) \times (\mathbb{M}_1, \ldots, \mathbb{M}_n) \to \mathbb{R}$ *is defined as*

$$d_{\text{WPM}}\big((x_1, \ldots, x_n), (y_1, \ldots, y_n)\big) = \Big( \sum_{i=1}^n w_i \cdot d_i(x_i, y_i)^2 \Big)^{\frac{1}{2}}, \tag{4}$$

*where* $w_i \in (0, +\infty)$, $x_i, y_i \in \mathbb{M}_i$ *for* $i \in 1, \ldots, n$, *is a metric on the space* $\mathbb{M}_1 \times \mathbb{M}_2 \times \cdots \times \mathbb{M}_n$.

The $d_{\text{WPM}}$ aggregates different data modalities present in the Cartesian Product structure while satisfying the metric properties. When all spaces $\{\mathbb{M}_i\}_{i=1}^n$ are the same, weights can be set to $\{w_i = 1\}_{i=1}^n$.

## 3.3 METRICS ON BAGS

Bags pose a unique challenge due to their varying size and the assumption of being permutation invariant. They can be seen either as *sets* (Nguyen et al., 2021), or *multisets* (Chuang & Jegelka, 2022), where the latter is more general (Xu et al., 2019). Let's denote Bags bold-faced $\mathbf{x} = [\![x_i]\!]_{i=1}^{n_x}$ and $\mathbf{y} = [\![y_j]\!]_{j=1}^{n_y}$ and their items normal-faced $[\![\cdot]\!]$ are used instead of usual $\{\cdot\}$ to emphasize that the bags can be multisets) Relating to Definition 2.1, we assume items $x_i$ to be of schema $\mathbb{M}$, $x_i \in \mathbb{M}$, and we denote we write $\mathbf{x} \in [\![\mathbb{M}]\!]$ for the bag.

A general formula for the Bag Metric is as follows:

**Definition 3.2** (Bag Metric ($d_{\text{BM}}$)). *Let $d$ be a distance between probability distributions on $\mathbb{M}$, $\alpha : \mathbb{N} \times \mathbb{N} \to \mathbb{R}$ be a non-negative non-zero function, $\beta \in \mathbb{R}^+$ and $d_c$ be a distance on $\mathbb{N}^+$, then $d_{\text{BM}} : \mathbb{M} \times \mathbb{M} \to \mathbb{R}$ is defined as*

$$d_{\text{BM}}(\mathbf{x}, \mathbf{y}) = \alpha(|\mathbf{x}|, |\mathbf{y}|) \cdot d(\mathbf{x}, \mathbf{y}) + \beta d_c(|\mathbf{x}|, |\mathbf{y}|), \tag{5}$$

*where $\mathbf{x} \in [\![\mathbb{M}]\!]$ and $\mathbf{y} \in [\![\mathbb{M}]\!]$, is a metric between* Bags *with items on the space $\mathbb{M}$.*

**Theorem 1.** $d_{\text{BM}}$ *is a multiset metric on* $[\![\mathbb{M}]\!]$.

The theorem is the consequence of Proposition 3.9 of Bolt et al. (2022). The term $\beta d_c(n, m)$ is needed for extending the distance on probability distributions to multisets.

Different settings of $d$, $\alpha$, and $\beta$ instantiates different distances of prior art. Fixing $d$ to a Wasserstein distance, $d_{\text{WA}}$, we obtain *Earth mover's distance* popular on 3D point clouds (Nguyen et al., 2021) for $\alpha = 1$ and $\beta = 0$,; *Unnormalized Wasserstein distance* used in Chuang & Jegelka (2022) to define pseudometric on trees pseudometric for $\alpha(|\mathbf{x}|, |\mathbf{y}|) = \max(|\mathbf{x}|, |\mathbf{y}|)$ and $\beta = 0$; *Earth mover's distance with cardinality comparison* (Bolt et al., 2022) for $\alpha(|\mathbf{x}|, |\mathbf{y}|) = \tau$ and $\beta = 1 - \tau$.

In Table 2, we present a list of distances on probability distributions used in our experiments, along with one widely recognized pseudo-distance. The theoretical foundations and formulas

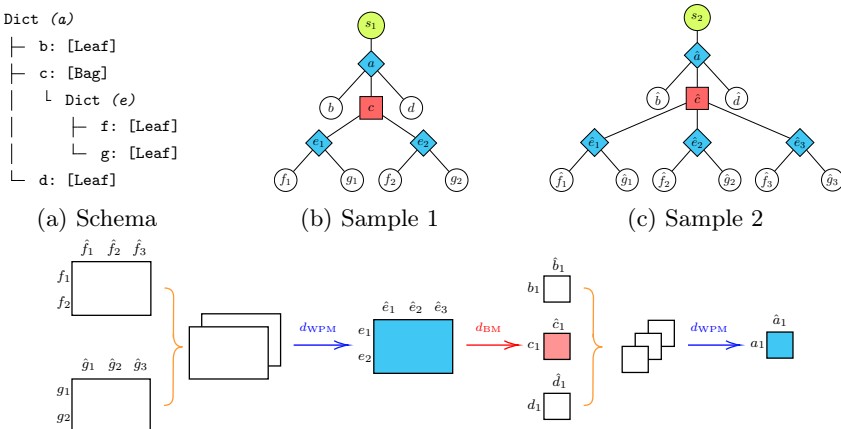

```
Dict (a)
├─ b: [Leaf]
├─ c: [Bag]
│   └ Dict (e)
│       ├─ f: [Leaf]
│       └─ g: [Leaf]
└─ d: [Leaf]
```

(a) Schema      (b) Sample 1      (c) Sample 2

(d) Schematics of the computation of the HTD between two samples.

Figure 2: Example of computation of the HTD between Sample 1 (b) and Sample 2 (c), both having schema shown in Subfigure (a). The computation goes bottom up, starting by computing pairwise distances between Leaves $\{f_1, f_2\}$ and $\{\hat{f}_1, \hat{f}_2, \hat{f}_3\}$, and $\{g_1, g_2\}$ and $\{\hat{g}_1, \hat{g}_2, \hat{g}_3\}$. Since they are children of a Dict $e$, the distance between nodes $\{e_1, e_2, \hat{e}_1, \hat{e}_2, \hat{e}_3\}$ is computed using $d_{\text{WPM}}$. Nodes $c$ and $\hat{c}$ are Bags; therefore, $d_{\text{BM}}$ is used to compute the distance between them utilizing the previously computed distance on nodes $e$. The computation is completed by computing distances between Leaves $b$ and $\hat{b}$, and $d$ and $\hat{d}$, which are then combined together with the distance between $c$ and $\hat{c}$ using $d_{\text{WPM}}$ resulting in the final distance on Sample 1 and Sample 2.

for these distances are provided in Appendix B. These distances are not the only ones available, so we refer the reader to Mroueh et al. (2017) for an extensive overview.

### 3.4 DISTANCE ON HS-TREE

The above distances defined on Dicts and Bags did not make any assumptions on the set of child items except that there exists a distance. This generality is important for the recursive definition of distance on HS-Trees.

**Definition 3.3** (HS-Tree distance (HTD)). *Let $\mathbb{H}$ be an arbitrary fixed schema of HS-Trees as defined in 2.1. Then the HTD distance $d_{\mathbb{H}}$ on $\mathbb{H}$ is defined recursively:*

1. **Leaves**: *If $\mathbb{H}$ is a leaf, then distance $d_{\mathbb{H}}$ is defined by a distance for the appropriate data type (see Section 3.1).*
2. **Bags**: *If $\mathbb{H} = [\![\mathbb{I}]\!]$ is a Bag, then the distance $d_{\mathbb{H}}$ is defined by a distance on (multi-)sets (see Section 3.3) with distance on items $d_{\mathbb{I}}$ being defined according to schema $\mathbb{I}$.*
3. **Dicts**: *Let $\mathbb{H} = \{(k_i, \mathbb{S}_i)\}_{i=1}^m$ be Dict. Then the distance $d_{\mathbb{H}}$ is as a distance on product of spaces (see Section 3.2), where distances $d_{\mathbb{S}_i}$ on sub-spaces are defined according to schemas $\{\mathbb{S}_i\}_{i=1}^m$.*

**Theorem 2.** *Let $\mathbb{H}$ be an arbitrary fixed schema from HS-Trees. Then an HTD distance exists on $\mathbb{H}$.*

The theorem is a consequence of the recursive definition. Formal proof is in Appendix D.

**Example:** The computation of HTD on samples from the Mutagenesis dataset is illustrated in Fig. 2. The computation starts by computing all pairwise distances between Leaves (white circles) with the same path to the root. Then, the computation progresses towards the root, using either distance on Bags or Dicts according to the type of inner nodes.

The computational complexity and theoretical properties of HTD mainly depend on the schema and the chosen distance function for the Bags. The universal choice is to use metrics for multisets, but this can be computationally expensive due to the need to compare

distributions (see Appendix B). Many times, especially when items of Bags have infinite support (e.g., one of their leaves is real), the probability that the set is multiset can be zero, in which case computationally cheap distances for sets (e.g., Haussdorf) are sufficient.

Let assume two samples in Fig. 2 with $c_b$, $c_f$, $c_g$, and $c_d$ being complexities of distances on leaves and $|f|$ denoting the size of Bags (which are assumed to be of equal size.) Assuming the complexity of distance on Bags being cubic, the complexity of distance on samples is $\mathcal{O}(|f|^2(c_f + c_g) + |f|^3 + c_b + c_d)$. The example is worked in detail in Appendix E.

## 4 RELATED WORK

Tree-edit distance (Zhang & Shasha, 1989) (TED) quantifies structural dissimilarity between rooted trees by calculating the minimum edit operations required for transformation. TED's applications range from computational biology to natural language processing (Sidorov et al., 2015). TED has been extended to heterogeoenous trees (Bille, 2005) and to HS-Trees in Šopík & Strenáčik (2022). Tree-edit distances are non-differentiable, and their computational complexity is cubic (Demaine et al., 2009).

A pseudo-distance for rooted homogenous trees (TMD) with fixed depth was proposed in Chuang & Jegelka (2022) to study properties of GNNs since the computational graph of GNNs equals to a tree (Xu et al., 2019; Errica & Niepert, 2023). The drawbacks of TMD are that it does not allow heterogeneous inner nodes and Leaves, it is not a proper distance, and its computational complexity is cubic. Interestingly, TMD implicitly uses product metric (4) with weights $w = 1$ and $\mathrm{L}_1$ distance to combine distance on features of the node with that of the neighborhood. The HS-Trees formalism makes it explicit that TMD is a special case of HTD for homogeneous graphs, using a different product metric and fixed weights.

Tree Kernels (Culotta & Sorensen, 2004; Schölkopf et al., 2004) transforms the tree structures into strings, which are then compared by String Kernels (Lodhi et al., 2002) similar to the Levenshtein distance. Kernels for sets viewed as samples from probability distributions have been proposed (Gretton et al., 2005).

Several methodologies emerged for supervised learning on rooted trees (Tai et al., 2015; Cheng et al., 2018; Socher et al., 2011; Mandlík et al., 2022; Woof & Chen, 2020), DAGs (Thost & Chen, 2021), and sets (Zaheer et al., 2017), but none of them is using distance. Recently, sum-product networks have been extended to HS-Trees (Papez et al., 2024), offering a tractable probabilistic model.

## 5 EXPERIMENTS

The experiments are designed to show the properties of the proposed HTD. On classification problems, we demonstrate the advantage of *Differentiability*, *Modularity*, and flexibility due to *Free Parameters*. On the anomaly detection task, we again demonstrate the advantage of flexibility. Finally, we demonstrate the advantage of the HTD for analysis of GNNs for heterogeneous graphs as opposed to homogenization suggested in Chuang & Jegelka (2022). We aimed to compare the methods under the same conditions and criteria. The implementation of HTD is available at `https://anonymous.4open.science/r/HSTreeDistance1`, and experiments are available at `https://anonymous.4open.science/r/HTDExperiments`.

The experiments use eight datasets, consisting of six hierarchically structured datasets sourced from Motl & Schulte (2015) (Mutagenesis, Hepatitis, Chess, Genes, Webkp, and Cora) and two datasets (MUTAG and BZR) sourced from Morris et al. (2020). Some datasets were originally graph datasets that were converted to tree-structured data. MUTAG and BZR were transformed by reproducing methods of Chuang & Jegelka (2022). The difference between Mutagenesis (Mut.) and MUTAG is that MUTAG is homogeneous, whereas Mutagenesis is heterogeneous with additional features on Leaves.

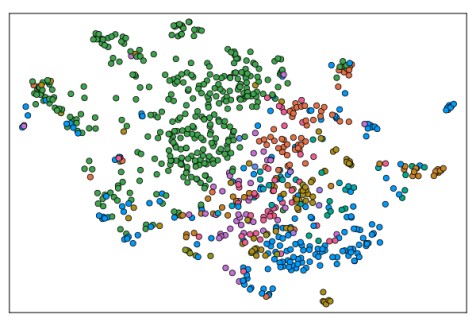 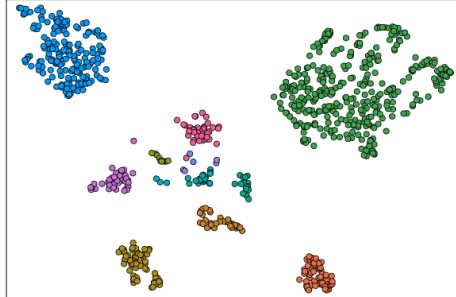

Figure 3: UMAP projection of Genes dataset using HTD with default parameters on the left and optimized parameters on the right. The total number of adjustable parameters is 21.

Table 3: Classification experiment results. Results are reported using an accuracy score. For datasets, MUTAG and BZR homogeneous graphs were unrolled to trees up to depth L=4.

| method | Mut. | Hepatitis | Chess | Genes | Webkp | Cora | MUTAG | BZR |
|---|---|---|---|---|---|---|---|---|
| HMIL | 87.8 | **92.5** | 41.5 | 98.8 | 82.0 | **85.3** | 91.0 | 88.2 |
| kNN-TED | 86.5 | 64.0 | 36.4 | 44.2 | 46.0 | 27.3 | 87.7 | 83.5 |
| kNN-TMD | — | — | — | — | — | — | 87.7 | 84.7 |
| SVM-TMD | — | — | — | — | — | — | 92.2 | 87.6 |
| kNN-HTD | **96.4** | 92.3 | **52.5** | **100** | **86.1** | 85.2 | 92.8 | **91.8** |
| SVM-HTD | **96.4** | 92.3 | 48.6 | **100** | 85.3 | 80.6 | **93.7** | 89.8 |
| GP-HTD | 91.9 | 84.3 | 41.2 | 93.6 | – | – | 75.7 | 87.3 |

## 5.1 DISTANCE-BASED CLASSIFICATION

This experiment compares the proposed HTD to tree-edit distance (TED) adapted to HS-Tree (Šopík & Strenáčik, 2022) and to the tree pseudo-distance (TMD) (Chuang & Jegelka, 2022), which is shown only on MUTAG and BZR as it requires homogenous trees. These distances are used with the k-Nearest Neighbor classifier, with the Support Vector Machine, and with the Gaussian Process. The HMIL classifier (Mandlík et al., 2022) based on neural networks is used as the baseline. All experiments were repeated five times. The best hyperparameters were selected according to accuracy on the validation set. Implementation and experimental details, together with a list of hyper-parameters, are provided in Appendix A. HTD treats the type of distance on Bags as hyperparameters, but weights of $d_{\mathrm{WPM}}$ distance on Dict are learned.

The results in Table 3 show that classifiers using the proposed HTD exhibit superior performance to other methods on almost all datasets. kNN-HTD and SVM-HTD have frequently performed better than HMIL classifier based on neural networks, but at the expense of higher complexity during classification due to naive implementation of the kNN classifier.

**Free parameters:** The good results of HTD are likely due to its flexibility introduced mainly by weights in the distance on Dicts (see Equation 6). This is supported by the fact that the Tree Mover's Distance (TMD), whose parameters were selected heuristically, is worse on MUTAG and BZR. The effect of good parameters is shown in Fig. 3 depicting distances between points of the Genes dataset with HTD with default parameters (all equal to one) and optimal parameters found by Contrastive learning (see below). An example of semi-supervised clustering is shown in Appendix F.

**Differentiability:** We compare three methods to optimize weights: random sampling, contrastive learning with Triplet loss (Weinberger & Saul, 2009), and kernel learning with the Gaussian Process. The last two methods require differentiability with respect to parameters. The results in Table 4 may suggest no significant difference between random sampling and contrastive learning, but contrastive learning yields better accuracy 25 times while random sampling is better only 13 times. Surprisingly, kernel learning with Gaussian Processes seems to be the least effective method.

Table 4: Classification performance with different methods of learning parameters for our HTD ($d_{\mathrm{HTD}}$). Some combinations were not evaluated due to stability issues or excessive computational demands of the method.

| b.m. | methods | model | Mut. | Hepatitis | Chess | Genes | Webkp | Cora | MUTAG | BZR |
|---|---|---|---|---|---|---|---|---|---|---|
| Hausdorff | RS | kNN | 91.9 | 86.7 | 49.1 | **100** | 51.3 | 31.7 | 81.1 | **91.8** |
| | | SVM | 94.6 | 90.0 | 40.1 | 99.8 | 47.6 | 32.6 | 79.3 | 89.3 |
| | Triplets | kNN | **96.4** | 84.7 | 49.1 | **100** | 52.8 | 32.9 | 84.7 | 90.1 |
| | | SVM | 94.6 | 83.7 | 39.6 | **100** | 53.0 | 33.5 | 79.3 | 89.7 |
| | GP | GP | 91.9 | 81.3 | 38.4 | 99.8 | – | – | 75.7 | 88.9 |
| Partial W. | RS | kNN | 91.9 | 86.7 | **52.5** | 98.8 | 49.9 | 74.6 | 91.9 | 89.7 |
| | | SVM | 92.8 | 88.3 | 42.4 | 67.9 | 47.9 | 63.8 | **93.7** | 85.6 |
| | Triplets | kNN | 95.5 | **92.3** | 48.6 | **100** | – | – | 92.8 | 89.2 |
| | | SVM | **96.4** | **92.3** | 43.0 | 99.4 | – | – | **93.7** | 89.3 |
| | GP | GP | 90.1 | 77.3 | – | – | – | – | 92.1 | 86.1 |
| Chamfer | RS | kNN | 91.9 | 90.0 | **52.5** | 97.0 | 72.0 | 84.3 | 83.8 | 89.3 |
| | | SVM | 90.1 | 90.3 | 43.5 | 86.9 | 82.0 | 80.6 | 84.7 | 89.7 |
| | Triplets | kNN | 94.0 | 86.0 | 46.4 | **100** | 86.1 | 70.1 | 82.0 | 88.6 |
| | | SVM | 93.0 | 74.3 | 42.4 | **100** | 85.3 | 80.6 | 86.5 | 89.8 |
| | GP | GP | 91.9 | 84.3 | 41.2 | 93.6 | – | – | 75.7 | 87.3 |

**Modularity:** The modularity of HTD is improved by selecting distances on Leafs and Bags. We kept those on Leafs fixed and explored three options on Bags: the Chamfer Distance, the Hausdorff Distance, and the Partial Wasserstein Distance. According to the results in Table 4, Partial Wasserstein performs overall the best, which is in line with theory as it is able to discriminate multisets. However, in all cases except one (MUTAG), the same accuracy can be achieved either by Haussdorff or Chamfer distance, which computational complexity scales quadratically instead of cubically (see Table 2). Only the MUTAG dataset contains categorical Leaves, which requires the recognition of multisets. Other datasets have at least one leaf with real values so distances on sets are, therefore, sufficient.

**Heterogenity:** Recall that the difference between Mutagenesis and MUTAG datasets is that the latter was homogenized as needed for the TMD distance (Chuang & Jegelka, 2022). Using the heterogeneous version with rich information in Leaves improves the accuracy by 3% and allows using the cheap Haussdorff with quadratic complexity. A similar experiment reported below on GNNs led to the same results.

## 5.2 Distance-based Anomaly detection

This section demonstrates the advantage of HTD in k-Nearest Neighbor anomaly detector, which provides a good baseline (Škvára et al., 2021). Since the HMIL classifier cannot be used for anomaly detection, it has been excluded from the experiments. Contrastive learning for tuning weights is impossible due to the lack of labels, but kernel learning with GP is possible. The experimental protocol mirrored that for the classification tasks with few modifications needed to adapt the datasets for the specific anomaly detection task. We accomplished this by following the leave-one-in procedure described in Škvára et al. (2021). The evaluation metric was also changed from accuracy to the AUC, which is usual in the anomaly detection community.

Table 5: Anomaly detection experiment result. Results are reported using the AUC score.

| method | Mut. | Hepatitis | Chess | Genes | Webkp | Cora | MUTAG | BZR |
|---|---|---|---|---|---|---|---|---|
| kNN-TMD | – | – | – | – | – | – | 86.1 | 71.3 |
| kNN-TED | 82.8 | 75.8 | 80.1 | 81.6 | 82.9 | 89.5 | 90.9 | 72.0 |
| kNN-HTD | **94.4** | **89.7** | 84.3 | **99.6** | **92.6** | **97.2** | 86.6 | 73.4 |
| GP-HTD | 90.0 | 78.5 | **84.4** | 96.4 | – | – | **91.9** | **75.1** |

The average AUCs from five repetitions are presented in Table 5. The dominance of kNN-HTD observed above is consistently replicated with few exceptions. TED works well on the MUTAG dataset where kNN-TED performs on par with other methods. GP-HTD shows some promises, but it is difficult to optimize without collapsing the optimization procedure. Random sampling of weights with kNN is a faster and more reliable method.

Table 6: Experimental results on heterogeneous and homogenized IMDB dataset.

(a) The correlation coefficient (multiplied by 100) between distances and GNNs.

|  | homo-HTD | hetero-HTD |
|---|---|---|
| homo-GNN | **53 ± 0.2** | 50 ± 1.4 |
| hetero-GNN | 52 ± 0.9 | **58 ± 0.2** |

(b) The F1-score (macro) of GNN classifiers.

|  | F1-score |
|---|---|
| homo-GNN | 67.2 ± 1.6 |
| hetero-GNN | 70.9 ± 1.1 |
| HetSANN (Hong et al., 2020) | 72.0 ± N/A |
| MAGNN-AC (Jin et al., 2021) | 60.8 ± N/A |
| SeHGNN (Yang et al., 2023) | 67.1 ± 0.3 |
| Simple-HGN (Lv et al., 2021) | 63.5 ± 1.4 |
| HTD - SVM | 67.1 ± 0.6 |

## 5.3 Analysis of Graph Neural Networks

Tree-Movers Distance (TMD) has been introduced in Chuang & Jegelka (2022) to study the stability of homogeneous GNNs (Hamilton et al., 2018) by showing that their output correlates with the TMD distance, estimated as $\text{corr}\left(\{\|\text{gnn}(v_i) - \text{gnn}(v_j)\|_2, d(v_i, v_j)\}_{i,j=1}^{1000,1000}\right)$ where $\text{gnn}(v)$ is embedding of vertex $v$ provided by GNN and $d(v_i, v_j)$ is a distance between computation trees of vertices $v_i$ and $v_j$. The proposed HTD enables us to extend this analysis to heterogenous GNNs since it adapts to their computational graph better than the homogenization suggested in Chuang & Jegelka (2022).

We measure the correlation on IMDB dataset (Fu et al., 2020), which is a heterogeneous graph with three types of nodes (actors, directors, movies) and four types of edges. The goal to predict the type of the actor based on its neighborhood is solved by heterogeneous (Zhang et al., 2019) and homogeneous GNNs. The homogeneous variant of IMDB was created by method from PyTorch geometric (Fey & Lenssen, 2019), which corresponds to the method recommended in Chuang & Jegelka (2022).

To demonstrate that the GNNs used for correlation analysis are well trained, Table 6b shows an F1-score on the testing set. The table also shows scores of other prior art (Hong et al., 2020; Jin et al., 2021; Yang et al., 2023; Lv et al., 2021) and for curiosity, an SVM classifier with the proposed distance used in Section 5.1. The results show that heterogeneous GNNs perform better than their homogeneous counterparts. Surprisingly, SVM with 6 parameters for metric and 3600 of SVM multipliers performs frequently better than GNNs with orders of magnitude more parameters.

The correlation coefficient between GNNs on heterogeneous and homogenized graphs and HTD on heterogeneous and homogenized trees is shown in Table 6a. As expected, the highest correlation occurs when the type of distance matches the type of computation graph. Specifically, the heterogeneous/homogeneous tree distance correlates with heterogeneous/homogeneous GNNs, and the correlation decreases in the case of mismatch.

## 6 Conclusion

This paper introduced Hierarchically-Structured Tree Distance (HTD), measuring the distance between samples emerging from popular data storage formats (e.g., JSON, XML, and ProtoBuffer) and naturally representing message passes in heterogeneous GNNs. We have demonstrated that this distance, paired with well-known distance and kernel-based algorithms, can solve common machine learning tasks like classification, anomaly detection, visualization, and clustering with performance frequently better than the state-of-the-art methods based on neural networks with orders of magnitude more parameters. A good performance of HTD is owed to its flexible differentiable parametrization, which allows it to be optimized for a given problem by common metric-learning algorithms.

The HTD distance lays a foundation for future research of tree-structured data and the development of new generative and self-supervised methods. We envision the use of HTD as a reconstruction loss in variational and masked autoencoders, potentially leading to strong pre-training methods for HS-Trees, which might be important for industry storing data in structured formats.

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

## A  IMPLEMENTATION DETAILS

This section provides details about methods used in our experiments for parameter learning. Specifically, the parameters we learn are the weights from $d_{\text{WPM}}$. We outline the specifics of each method and how they were used in the study. Additionally, Tables 7 and 8 lists ranges of hyperparameters. These tables serve as a reference for understanding the settings and configurations used in our experiments. It's important to note that all experiments were repeated five times with different train/validation/test splits, where the validation splits were used for selecting the best hyperparameters, including HTD's parameters.

**Random Sampling**  The most naive but powerful method to set weights is random sampling (RS). Weights are sampled from a predefined distribution, and the best model (kNN and SVM) is selected according to the accuracy on the validation split. RS is inexpensive to compute for a single realization of weights as it requires no gradients. However, to find optimal parameters, hundreds or thousands of combinations (depending on the dataset) must be evaluated. For our study, we evaluated 500 random realizations of weights for each dataset.

**Contrastive Learning**  The second approach used contrastive learning, which uses labels and needs the distance function to be differentiable. We use the usual triplet loss (Weinberger & Saul, 2009), $\mathcal{L}_{tr}$, as

$$\mathcal{L}_{tr} = \max(d(x_a, x_p) - d(x_a, x_n) + 1, 0) + \beta \cdot \|\theta - \gamma\|_2,$$

where $x_a, x_p, x_n$ are the anchor, positive, and negative samples, respectively, and *theta* represents the parameters of the metric $d$.

The process of sampling triplets $x_a, x_p, x_n$ for minibatches is often more important than selecting appropriate values for $\beta$ and $\gamma$. To address this, we explored three different strategies: Random, Batch Hard, and Alternating.

1. **Random**: This method selects an anchor randomly, and then, depending on its label, positive and negative samples are again sampled randomly.

2. **Batch Hard**: first samples a subset (batch) of samples and computes the pairwise distance using the distance function $d$ with the current $\theta$. Then, it randomly samples the anchor and creates the most difficult triplet by sampling the most distant positive sample and the closest negative samples. This method encourages the distance function $d$ to distinguish the most challenging observations. However, it may fail in the presence of multiple clusters and outliers.

3. **Alternating**: alternates between the Random and Batch Hard methods.

**Kernel Learning with Gaussian Process**  Since HTD is a proper metric, it can be readily used in a kernel function applicable within Support Vector Machines (SVMs) or Gaussian processes. Optimization of kernel's parameters for SVM is difficult to optimize, but it is simpler in Gaussian Processes.

Gaussian Process acts as a probability distribution over training dataset $\mathbf{x} = \{x_1, \ldots, x_n\}$

$$GP(\mathbf{x}) = \mathcal{N}\Big(m(\mathbf{x}), K(\mathbf{x}, \mathbf{x})\Big),$$

where $m(\mathbf{x})$ is mean function and $K(\mathbf{x}, \mathbf{x})$ is covariance (kernel) matrix

$$K(\mathbf{x}, \mathbf{x}) = \begin{bmatrix} k(x_1, x_1) & \ldots & k(x_1, x_n) \\ \vdots & \ddots & \vdots \\ k(x_n, x_1) & \ldots & k(x_n, x_n), \end{bmatrix}$$

with respect to kernel function $k$, which we define as $k(x, y) = \exp(-d(x, y))$.

This means that when maximizing the likelihood of a Gaussian process, we can compute gradients not only with respect to the covariance matrix $K$ or the kernel function $k$ but also with respect to the parameters of the distance function $d$. This enables us to use this approach to learn the parameters of $d$.

Table 7: General hyperparameters for HTD.

|  | hyperparameter | value set |
| --- | --- | --- |
| General | Numeric Leaf metric | $L_2$-norm |
|  | Categorical Leaf metric | $L_2$-norm$/\sqrt{2}$ |
|  | Dict metric | $d_{\text{WPM}}$ |
|  | Bag metric | $d_{\text{CH}}, d_{\text{PW}}, d_{\text{WA}}, d_{\text{HA}}$ |
| Random Sampling | $d_{\text{WPM}}$ weight distribution | $0.35 \cdot \mathcal{U}(0, 100) + 0.65 \cdot \exp(9)$ |
|  | number of repetitions | 500 |
| Contrastive Learning | triple selection | {Random, Batch Hard, Alternating} |
|  | optimiser | Adam |
|  | learning rate | $\{1.0, 0.1, 0.01, 0.001\}$ |
|  | $\beta$ | $\{0, 0.1, 0.001, 0.001, -0.001\}$ |
|  | $\gamma$ | $\{0, 1\}$ |
| Kernel Learning | kernels | {Laplacian, Gaussian, Matérn32} |
|  | optimiser | {Adam, L-BFGS} |
|  | learning rate (for Adam) | $\{1.0, 0.1, 0.01, 0.001\}$ |

Table 8: Hyperparameters and their ranges of HMIL, kNN, SVM and GP classifiers.

| model | parameter | value set |
| --- | --- | --- |
| General | (CLF) split ratios | 60%/20%/20% (train/valid/test) |
|  | (AD) split ratios | 60%/20%/20% of *normal* data |
|  |  | 0%/50%/50% of anomalies |
| HMIL classifier | maximum epochs | $\{1000\}$ |
|  | number of neurons | $\{10, 20, 30, 40, 50, 60, 70, 80, 90, 100\}$ |
|  | aggregation function | $\{\text{mean}, \text{max}, \text{mean} + \text{max}\}$ |
|  | activation function | $\{\text{identity}, \text{relu}\}$ |
|  | batch size | $\{16, 32, 64\}$ |
|  | learning rate | $\{0.001, 0.0005, 0.0001\}$ |
|  | early stopping criterion | validation accuracy |
|  | early stopping patience | $\{30\}$ |
| kNN | number of neighbors | $\{1, \ldots, 150\}$ |
| SVM | $\gamma^{-1}$ | $\{0.1, 0.2, 0.3, \ldots, 20.0\}$ |
|  | kernels | $\{\text{Laplacian}\} \exp(-\gamma \cdot d(x, y))$ |

## B  DISTANCES FOR PROBABILITY DISTRIBUTIONS

In this section, we list distances on probability distributions and sets and discuss their theoretical properties, computational complexity, and underlying assumptions.

**Wasserstein Distance:** *Let* $(\mathbb{M}, d)$ *be complete and separable metric space, then for* $p \in [0, +\infty]$ *the p-Wasserstein distance between probability measures* $P_X$ *and* $P_Y$ *on* $\mathbb{M}$ *with finite p-moments is defined as*

$$d_{\text{WA}}(P_X, P_Y) = \inf_{\gamma \in \Pi(P_X, P_Y)} \left( \mathbb{E}_{(X,Y) \sim \gamma}[d(X, Y)^p] \right)^{1/p}, \tag{6}$$

*where* $\Pi(P_X, P_Y)$ *is set of all joint probability distributions* $\gamma$ *on* $\mathbb{M} \times \mathbb{M}$, *whose marginals are* $P_X$ *and* $P_Y$.

The Wasserstein distance (Panaretos & Zemel, 2019) is a popular distance between two probability distributions defined on the same metric space. It is applied to Bags by treating them as two empirical distributions. The distance is popular in image processing, natural language processing (NLP), and point cloud generation, where it is called Earth Mover's Distance ($d_{\text{EMD}}$) (Andoni et al., 2008) or Mallows distance (Levina & Bickel, 2001). The computational complexity of Wasserstein distance between two Bags $\mathbf{x}$ and $\mathbf{y}$ is cubic when $|\mathbf{x}| = |\mathbf{y}|$.

**Partial Wasserstein Distance:** *Let* $\mathbf{x} = [\![x_i]\!]_{i=1}^{n_x}$ *and* $\mathbf{y} = [\![y_j]\!]_{j=1}^{n_y}$ *be two multisets and* $d_{\text{WA}}$ *is Wasserstein Distance. W.L.O.G. assume that* $n_x > n_y$, $n_0 = n_x - n_y$, *then the Partial Wasserstein Distance between is defined as* $d_{\text{PW}}(\mathbf{x}, \mathbf{y}) = d_{\text{WA}}(\mathbf{x}, \mathbf{y} \cup [\![\phi]\!]_{k=1}^{n_0})$, *where* $\phi$ *is a special "null element" whose distance to other elements is maximum, i.e.,* $d(\phi, x) > d(y, x), \forall x, y \in \mathbb{M}$.

(a) Example of JSON file.  (b) Mutagenesis dataset scheme.

Figure 4: An example of a JSON file and the scheme of the (simplified) Mutagenesis dataset (Cheplygina & Tax, 2015). The term *Leaf* represents nodes with elementary data types, while Bag and *Dict* is equal to *Array* and *Object*, respectively.

Partial Wasserstein Distance is tailored to multisets (Chapel et al., 2020; Chuang & Jegelka, 2022) because the Wasserstein distance cannot differentiate between two multisets containing identical elements with different cardinalities, for example, $x = [\![1, 2, 1, 2]\!]$ and $y = [\![1, 2]\!]$. This problem is pertinent to all metrics designed for probability distributions and sets (which, by definition, removes the duplicates). The Partial Wasserstein Distance solves the problem by extending the space $\mathbb{M}$ where elements of the Bag live with a special *null element*, $\phi$. When computing a distance, a null element is used to equalize the cardinality. In the above example, this would correspond to transforming $\mathbf{y}$ into $\mathbf{y}' = [\![1, 2, \phi, \phi]\!]$.

$d_{\mathrm{PW}}$ with $\alpha(n, m) = \max(n, m)$ and $\beta = 0$ in Equation (5) is used in (Chuang & Jegelka, 2022) to define pseudo-distance on homogeneous trees with fixed depth, used to analyze graph neural networks with sum aggregation function. $d_{\mathrm{PW}}$ is used in this paper exclusively with this setting.

**Hausdorff Distance:** *Let* $\mathbf{x} = [\![x_i]\!]_{i=1}^{n_x}$ *and* $\mathbf{y} = [\![y_j]\!]_{j=1}^{n_y}$ *be two sets of points in some metric space* $(\mathbb{M}, d)$*, the Hausdorff distance (Huttenlocher et al., 1993) is computed as*

$$d_{\mathrm{HA}}(\mathbf{x}, \mathbf{y}) = \max \big\{ \max_i \min_j d(x_i, y_j), \max_j \min_i (x_i, y_j) \big\}. \tag{7}$$

Hausdorff distance measures the similarity between two sets of points in a metric space. It is used in image analysis, shape recognition, and pattern matching. The naive implementation has quadratic complexity, but efficient algorithms (Taha & Hanbury, 2015) with linear complexity in expectation exist.

**Chamfer (pseudo-) Distance:** *Let* $\mathbf{x} = [\![x_i]\!]_{i=1}^{n_x}$ *and* $\mathbf{y} = [\![y_j]\!]_{j=1}^{n_y}$ *be two sets of points in some metric space* $(\mathbb{M}, d)$*, then Chamfer pseudo-distance (Huttenlocher et al., 1993) is computed as*

$$d_{\mathrm{CH}}(\mathbf{x}, \mathbf{y}) = \frac{1}{|\mathbf{x}|} \sum_i \min_j d(x_i, y_j) + \frac{1}{|\mathbf{y}|} \sum_j \min_i d(x_i, y_j). \tag{8}$$

Chamfer pseudo-distance (Borgefors, 1988) (violates triangular inequality) also measures similarity between two sets of points as the minimum cumulative distance needed to transform one point set into another. Its computational complexity is quadratic, which makes it a more popular option than expensive Wasserstein distance.

## C  RELATION OF HS-TREES AND JSONS

This section uses the JSON format as a prototypical example of hierarchical formats. Data in JSON format are stored by combining elementary data types: *Strings, Numbers, Booleans,* and *Null* with two composite data types: *Objects* and *Arrays*.

The *elementary data types* corresponds to Leaf in HS-Trees, as they do not have children. Distances for these data types exist and are discussed in Section 3.1.

*Object* is a key-value dictionary in which the key is restricted to *String*, and values can be any JSON data type. Keys must be unique and serve as a semantic data identifier in the corresponding value field. *Object* therefore corresponds to Dict in HS-Trees.

*Array* is a sequence of elements of any JSON data type with arbitrary length. In general, items of the array do not have to be of the same data type (schema in the terminology of this paper), but we impose this restriction as it is very common in the industry. This work also assumes the data in arrays to be unordered, which can be restored using position encoding. With that, *Arrays* maps to Bag in HS-Trees.

For example, in Fig. 4a, the value corresponding to the key `bonds` is a Bag. Values in this bag are Dicts. The first dictionary in this array contains key-value pairs `"element":    "c"`, and `"charge":0.2`. The values corresponding to these keys: `"c"` and `"0.2"` are Leaves, as they are elementary types.

## D    Proof of Theorem 2

**Theorem 2.** *Let $\mathbb{H}$ be an arbitrarily fixed schema of HS-Trees. Then, an HTD distance exists in $\mathbb{H}$.*

*Proof.* Due to the recursive construction of the set of all schemas, the proof is carried by induction.

Let $\mathbb{H}$ be a schema with depth 0, which means that all samples $x \in \mathbb{H}$ are trees with depth 0, i.e. they are Leaves. Then $\mathbb{H}$ is an elementary data type, and the distance can be computed by an appropriate choice listed in Section 3.1.

Carrying the induction, we now assume to be able to define distance for all schemas of depth $l - 1$, and we want to define distance on samples of schema $\mathbb{H}$ with depth $l$, where $l > 0$. Let $x, y \in \mathbb{H}$, then the top-node of $x$ and $y$ is either of type Bag or Dict.

Let's first assume roots to be Bags and denote $\{x_i\}_{i=1}^n$ and $\{y_j\}_{j=1}^m$ items (childs) of $x$ and $y$ respectively. By definition of HS-Trees and their schema 2.1, all items $x_i$ and $y_i$ have the same schema $\mathbb{I}$ with a depth $l - 1$, and by induction assumption, there exists a distance $d_{\mathbb{I}}$. The existence of $d_{\mathbb{I}}$ is sufficient to define a distance between (multi)-sets as discussed in Section 3.3.

Alternatively, roots are Dict. Then $x = \{(k_i, x_i)\}_{i=1}^n$ and $y = \{(k_i, y_i)\}_{i=1}^n$, where $\{k_i\}_{i=1}^n$ are unique keys and $x_i, y_i$ are corresponding values with the schema $\mathbb{S}_i$ of length at most $l - 1$. By induction principle, there exist distances $d_{\mathbb{S}_i}$ on $\mathbb{S}_i$, which are the sufficient condition to define distance between $x, y$ using for example the weighted product metric discussed in Section 3.2. $\qquad\square$

## E    Example of computing the complexity

Let's demonstrate the complexity of the distance between two samples in Fig. 2. For simplicity, we assume that Bags (labeled $c$) have the same length. The computation starts by computing distances between Leaf nodes $f$ and $g$. Although $L_2$ metric has linear complexity with respect to its dimension, denoted by $c_f$ and $c_g$, the Bag distances requires to compute all pairwise distances and therefore the complexity with respect to their number, i.e. $\mathcal{O}(|f|^2(c_f + c_g))$, where $|f|$ represents the number of Leaves with label $f$. The complexity of the $d_{\text{WPM}}$ for $e$ is the sum of complexities of leaves, which was already included. The complexity of computing distance on Bags of node with label $c$ is either cubic, which makes the complexity $\mathcal{O}(|f|^2(c_f + c_g) + |f|^3)$, or quadratic which yields complexity $\mathcal{O}(|f|^2(c_f + c_g + 1))$. To finish the computation of distance, we need to compute the distance of Leaves $b$ and $d$ and add it to the distance $c$. The final complexity is therefore $\mathcal{O}(|f|^2(c_f + c_g) + |f|^3 + c_b + c_d)$ or $\mathcal{O}(|f|^2(c_f + c_g + 1) + c_b + c_d)$.

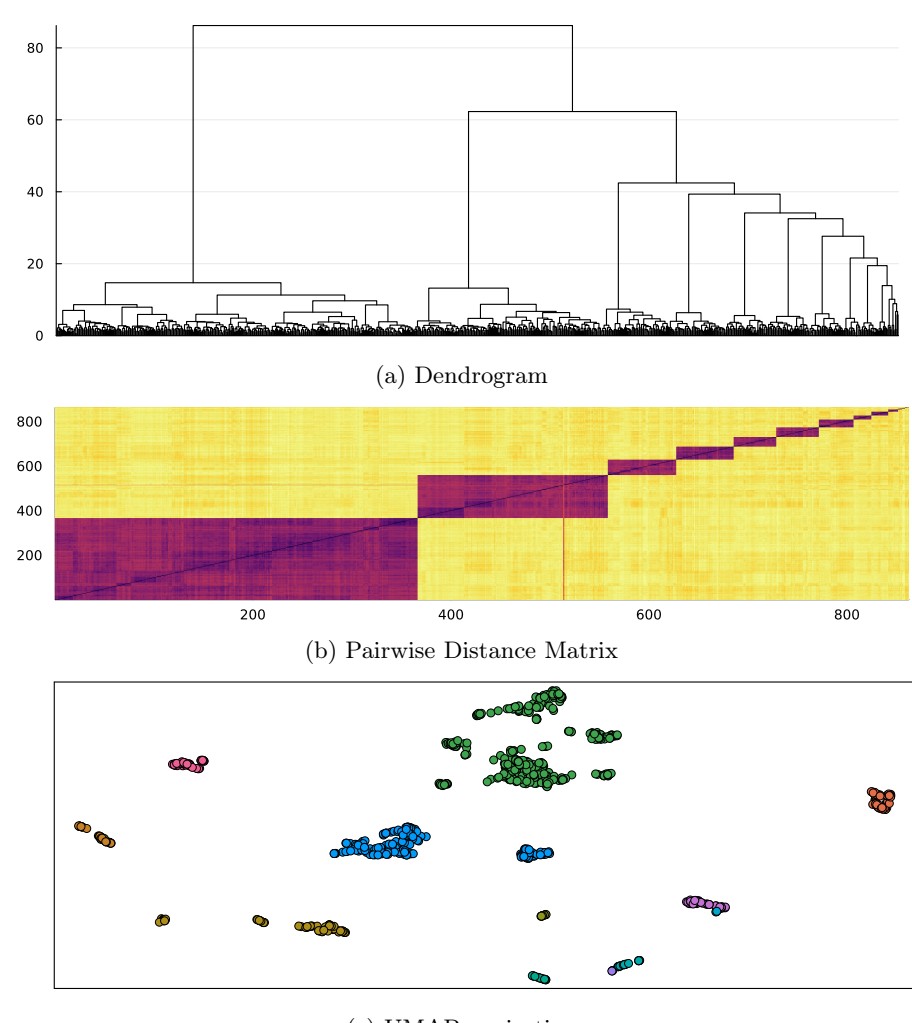

(a) Dendrogram

(b) Pairwise Distance Matrix

(c) UMAP projection

Figure 5: Visualization of hierarchical clustering results on the *Genes* dataset using (a) Dendrogram, which represents the clustering structure based on pairwise distances, (b) Pairwise Distance Matrix heatmap, and (c) UMAP projection colored by labels. The distance matrix is computed using HTD and ordered to match the branches of the dendrogram.

## F    SEMI-SUPERVISED CLUSTERING

In this experiment, we demonstrate that HTD can also be effectively used for clustering. We opted for a semi-supervised clustering approach because, as shown in Section 5, learning optimal parameters allows HTD to better fit the dataset. Initially, a randomly selected 20 percent of the labeled data was used to learn the parameters through contrastive learning, after which the pairwise distance matrix (PDM) for the entire dataset was computed. Many clustering algorithms can then be applied directly once the PDM is available. Figure 5a shows the dendrogram produced by hierarchical clustering on the Genes dataset. The results indicate two large clusters along with several smaller ones, consistent with the UMAP projection of this dataset shown in Figure 5c and computed using the same PDM.

