# OpenReview forum: "Differentiable Distance Between Hierarchically-Structured Data"
_ICLR.cc/2025/Conference — Submitted to ICLR 2025_

### Official Review · Reviewer_ehnU · 2024-10-29

**Soundness:** 2
**Presentation:** 1
**Contribution:** 2
**Rating:** 3
**Confidence:** 3

**Summary:**

The paper introduces hierarchically-structured tree distance (HTD) between HS-Trees.

**Strengths:**

The authors introduce a new distance for hierarchically-structured trees based on Leaves, Bags, and Dicts.

**Weaknesses:**

- The motivation and explanation of the use of HS-trees and their distance are unclear in the introduction. The authors first say that the properties of HT-Trees are used in existing work. Then, in the next paragraph, the authors mention that the distance on HS-Trees has been studied very little. Among the properties used in previous work, were there HS-Tress distance?
- In the introduction and background, it’s unclear if the term HS-Trees is from previous work or if the term is first given by the authors. In addition, the background of HS-Trees, sample, and schema are very confusing. It would be clearer if the authors could provide an example of what they are here. Moreover, it’s unclear how bags are assumed to be “permutation invariant; therefore, the position has to be encoded through position encoding” and how the “universal approximation theorem for HS-Trees has been proved in Pevny & Kovarik (2019).”
- The authors mention the motivation of the work is “measuring the distance between samples emerging from popular data storage formats (e.g., JSON, XML, and ProtoBuffer)”. However, in the experiment section, there is no application in such data format. It is misleading to use the data storage formats as motivation in the introduction, but there is no related experiment
- In Section 2.1, it is unclear if the authors are trying to claim that GNNs are “hierarchically-structure data”. In the context of GNN, it’s hard to see what sample, schema, and even HS-Trees are.
- The authors claim that the proposed distance is differentiable, however, there is no theoretical proof
- The term Leaves is commonly used in tree structure data. It will be important to differentiate the difference between common tree structures and HS trees. In addition, it’s unclear how and why the definition of Leaves in Section 3. 1 is outside of the scope
- There is no proof for Theorem 1. The justification right after Theorem 1 is hard to follow.
- It is hard to link the relation between Eq. (5) and the distance in Table 2, even with the brief introduction in Appendix B. A simple proof or derivation could have helped to understand the connection.
- It’s unclear what the relation between HTD and TMD is.  A simple proof or derivation could have helped to understand the connection.
- The README.md files in the provided link to the code are not sufficient to reproduce the HTD and experimental results.
- The experimental setup is unclear in the main texts. It’s unclear what the actual classification tasks and anomaly detection are. Also, while Appendix A includes the implementation details, there is no reference in the main text link to the appendix. In addition, in Appendix A, the authors claim that the experiments are repeated five times, and there is no variance or standard deviation reported in the performance in the main text.
- It’s unclear what the colors represent in Figure 3. It’s unclear why and what “—” represents in Tables 3-5.
- The paper needs more proofreading: i) additional ) in line 029, ii) line 250 missing a period and there is an additional ), iii) notation is very hard to follow; a notation is given with confusing comma

## Minor
- There if no reference and introduction when the Mutagenesis dataset is first mentioned in line 317

**Questions:**

- Only until Figure 2 the authors demonstrate examples of Schema. However, the sample 1 and sample 2 are essentially trees, it still remains unclear how HS-Trees to GNN in Section 2.1. Could the author provide an illustrative figure to show the relation?
- The font size is a bit bigger than usual?

---

### Official Review · Reviewer_su3h · 2024-10-29

**Soundness:** 2
**Presentation:** 3
**Contribution:** 2
**Rating:** 5
**Confidence:** 3

**Summary:**

This paper introduces the Hierarchically Structured Tree Distance (HTD), a metric designed to measure distances between tree-structured data commonly stored in formats like JSON and XML. HTD effectively represents message passing in heterogeneous graph neural networks (GNNs). Experimental results show that this distance metric is capable of addressing various machine learning tasks, including classification, visualization, and clustering.

**Strengths:**

- The paper is well-written.
- The proposed HTD generalizes the tree mover’s distance, making it applicable to both heterogeneous graphs and tree-structured data.
- Extensive experiments conducted across multiple tasks—classification, clustering, and anomaly detection—show HTD's superiority over state-of-the-art methods for tree-structured data.

**Weaknesses:**

I have several concerns regarding the novelty of the proposed distance:
- HTD appears to be a straightforward extension of the tree mover’s distance, replacing the optimal transport (OT) distance with the Hausdorff and Chamfer distances, which may introduce cheaper computational complexity.
- It is unclear why HTD outperforms the tree mover’s distance on homogeneous graph datasets, such as MUTAG and BZR, as shown in Table 3.
- Even on heterogeneous datasets, it seems feasible to apply the tree mover’s distance by constructing separate computational trees for each node type in the graph. So we can improve the performance of tree mover's distance on heterogeneous datasets like MUTAG and BZR.
- Similar to the tree mover’s distance, HTD does not meet the criteria for defining a valid kernel for tree-structured data, as it is not conditionally negative definite.
- Including standard deviations (STD) in the results of Tables 3 and 4 would be beneficial, as the variances are large; for instance, the STD of accuracy values for MUTAG and BZR might be around 5.

**Questions:**

See the weaknesses above.

---

> ### Author Response · Authors · 2024-11-19
> **Answer**
>
> Thank you for raising interesting points of the manuscript.
>
> * The proposed HTD can be indeed viewed as a generalization of TMD, but this generalization is not only due to more choices of metrics on bags (sets / multisets), but also through introduction of trainable weights in distances on dicts. Based on the experimental results, it seems that especially the latter (the parametrization in dicts) play the crucial role. For example, Figure 3 shows UMAP projection of distances on genes dataset, which both uses Hausdorff distance but differs in values of parameters in distances on dicts. A similar phenomenon can be seen in Table 4, where distances on bags (Hausdorff, Wasserstein, Chamfer) have typically smaller impact than how parameters on dicts are optimized (RS, triplets, GP).
>
> * The reason why HTD performs better than TMD on Mutag and BZR is again because of higher flexibility due to trainable weights in distances on dicts. TMD prescribes weights based on the level in the tree, whereas HTD adapts them for a given problem. Also notice that Hausdorff distance  seems to be more suitable for BZR dataset than partial Wasserstein (see Table 4), which is because BZR has continuous features and multisets might be rare.
>
> * We agree that TMD can be extended to heterogeneous trees, but the extension is not straightforward. Firstly, distances from different types of nodes has to be combined, likely through a weighted sum, which would correspond to our distances on dicts. Second, the separation to different types of trees might be in general difficult. Consider for example a sample, which in rootnode contains three types bags (type A, type B, and type C), and each element in each bag contains another set of three types of bag (e.g. Items of Type A contains bags if (type AA, type AB, tupe AC) and analogically for types B and C). Should we divide this into 9 types of homogeneous trees? This would certainly lead to combinatorial explosion, unlike in our construction.
>
> * *HTD does not meet the criteria for defining a valid kernel for tree-structured data, as it is not conditionally negative definite.*
>
>     This is a very good comment. This means that one has to be careful when using the distance inside RBF kernels with GP and SVM. In that case, we should treat the kernel as a noisy observation of the true positive (semi-)definite kernel. We have not observed any issues with numerical and convergence stability with SVM. Nevertheless, the GP required very peaky kernels like Laplace to ensure that the kernel matrix will be positively definite.
>
> * We have decided to present the results without them for the sake of brevity. We add them to the appendix.

---

> > ### Comment · Reviewer_su3h · 2024-11-25
> >
> > Thank the authors for the clarification. I'd like to keep my score unchanged.

---

### Official Review · Reviewer_cnv9 · 2024-11-04

**Soundness:** 2
**Presentation:** 3
**Contribution:** 2
**Rating:** 5
**Confidence:** 3

**Summary:**

This paper presents the Hierarchically-structured Tree Distance (HTD), a novel metric for structured data in formats like JSON and XML. Designed for rooted heterogeneous trees, HTD is modular and differentiable, allowing it to adapt to tasks like classification, clustering, and anomaly detection. Experiments show that HTD-based algorithms perform competitively with neural network methods while using far fewer parameters and are more effective for analyzing heterogeneous Graph Neural Networks than the Tree Mover’s Distance.

**Strengths:**

1. The paper is overall well-written and easy to follow.
2. This paper demonstrates theoretical superiority, as shown in Table 1.

**Weaknesses:**

1. The authors demonstrate the effectiveness mainly on distance-based tasks, which shows better performance compare to other distance-based method but does not appear comparable to GNN classifiers.

2. I am also concerned about the contribution and scope of this paper; however, I acknowledge that I am not an expert and am open to other opinions.

3. Although some limitations are mentioned in the submission (e.g., in the caption of Table 4), there is no comprehensive discussion of the proposed method's limitations.

**Questions:**

see weakness

---

> ### Author Response · Authors · 2024-11-16
> **Answers**
>
> The main contribution of the paper is to propose distance on data structures (hierarchically structured trees), which are ubiquitous in the industry. A mathematically correctly defined distance therefore allow to use many distance based algorithms on these types of data. The second important impact is that well-defined distance is important for mathematical analysis of algorithms and methods.
>
> With respect to the above, the goal was not create new algorithms competing with neural networks, but to define a metric. Neural networks have orders of magnitude more parameters and they are more flexible than the distance based algorithms.  It is therefore surprising that distance based algorithms with the proposed family of distances is frequently better than neural networks. In Table 3, k-nearest neighbor with the proposed distance is better on 5 out of 7 problems than state of the art neural network for this type of problems (HMIL). Importantly, HMIL is better by 0.2 and 0.1 percent, which is rather small. In Table 6b, we as a side-result compared the proposed distance on the movie dataset, where it was somewhere in the middle, but better than many graph neural networks utilizing sophisticated constructions like self-attention. These experimental results show that the distance is not only well mathematically founded, but works very well in practice.
>
> We acknowledge that we have not included paragraph about the limitations. We thought that we have made it sufficiently clear in the text. We list them clearly below and we happily list them in the manuscript, if it improves it.
>
> * The distance is parametrized by weights on individual sub-trees, which makes it flexible and can be therefore tailored to different problems. These weights also represent a problem of setting them which is nevertheless present in most other distances, even on Euclidean. There are therefore many algorithms and methods, how to optimize them. Since the proposed metric is differentiable, most of them can be used to optimize the proposed distance, which we have demonstrated in the paper In Table 4.
>
> * The second limitation is the computational complexity, which is determined mainly by the choice of the distance on bags. This can be for most general case of multi-sets cubic, but in many cases it is sufficient to used cheaper distances with quadratic complexity. This is described in detail a paragraph starting at line 322.
>
>
> Thank you for reviewing our paper and positive comments.

---

> > ### Comment · Reviewer_cnv9 · 2024-12-01
> >
> > Thank you for your responses. I will keep my score.

---

### Official Review · Reviewer_bcpr · 2024-11-06

**Soundness:** 3
**Presentation:** 3
**Contribution:** 3
**Rating:** 3
**Confidence:** 4

**Summary:**

The paper studies hierarichically structured data and introduces a tree-distance with differentiable parameters weighting the importance of different subspaces. The paper presents experimental evidence that their approach achieves similar performance to SOTA methods based on neural networks while having orders of magnitude fewer parameters, and also has some benefits for heterogeneous Graph Neural Networks compared to prior methods.

The paper is motivated by the fact that there are many structured data formats such as JSON/XML/Protobuffer but not a good way of defining a reasonable notion of distance between them, which is in contrast with what happens when we deal with more standard objects like vectors in Euclidean space.

This paper proposes a particular distance called HTD distance, which exploits the recursive nature of the previously-mentioned data formats. The ultimate goal is to have a modular construction by combining potentially different metrics on different levels of the given tree. HTD has weight parameters, which control importance on different parts, is differentiable, and requires orders of magnitude fewer parameters than neural networks with similar guarantees (based on experiments). The authors perform a series of experiments with supervised learning, ianomaly detection, heterogenous GNNs,  clustering and UMAP for visualization.

**Strengths:**

+the paper studies a natural problem on hierarchies which is how to define suitable metrics that are differentiable and modular.

+the authors present some natural candidate and apply it to different types of hierarchical data

+the authors present experimental results showcasing properties of their proposed metrics and benefits over prior methods.

**Weaknesses:**

-the theory is very straightforward in this paper. In fact, the two theorems stated as Th1 and Th2 could be obserations or propositions as they follow from the basic definitions.

-there have been recently approaches to define differentiable objectives suitable for doing optimization over trees and hierarchies, especially to deal with problems on relational data coming from networks (e.g. facebook or other social networks) with the goal of performing hierarchical clustering. The first such works were 1) Nickel et al. "Poincaré embeddings for learning hierarchical representations" and later 2) "Hyperbolic graph neural networks" of Nickel et al. and later the works of 3) Chami et al. "Hyperbolic graph convolutional neural networks" and 4) "From Trees to Continuous Embeddings and Back: Hyperbolic Hierarchical Clustering" and of 5) Monath et al. "Gradient-based hierarchical clustering using continuous representations of trees in hyperbolic space" have dealt with similar questions. I am surprised the authors do not cite such works as the problem of optimization over trees was addressed using differentiable methods in all of these works.

-omission of discussion for use of hyperbolic techniques and hyperbolic spaces in the present paper which is known to be suitable for hierarchical relations, much more than euclidean spaces.

**Questions:**

-Please I would like to hear the authors discussi the related works on Hyperbolic spaces for dealing with hierarchical data and Hierarchical clustering, where data also have latent tree structure. The goal here is to compare different trees and assign a loss to each so that lower loss means better tree for the dataset. The approaches there are also differentiable so how do you compare with them?

-For your metrics, is there any hope to prove something about the quality of the metric found?

---

> ### Author Response · Authors · 2024-11-14
> **Answers**
>
> The goal of our work is to define a distance on the space of hierarchically-structured trees. We thrived to have proper distance, since it is important for the theory behind many popular algorithms  and for the mathematical analysis.  The construction is on the end relatively straightforward, but we not certain it is obvious including the complexity.
>
> The experiments were intended to demonstrate the qualities of the distance when used inside common machine learning algorithms. We were pleasantly surprised how well it worked, as it is frequently better than state of the art methods with orders of magnitude more parameters (Table 3 and Table 6b). The experiments were not meant to propose a new state of the art on tested problems, the goal was to show that the proposed distance is not only mathematically sound, but also have good properties on practical problems.
>
> **I would like to hear the authors discussi the related works on Hyperbolic spaces for dealing with hierarchical data and Hierarchical clustering, where data also have latent tree structure. The goal here is to compare different trees and assign a loss to each so that lower loss means better tree for the dataset. The approaches there are also differentiable so how do you compare with them?**
>
> With all the respect, we do not understand how the work on hyperbolic spaces relates to the definition of a distance on HS-trees. Let us go in more details into individual works you have mentioned.
>
> * Nickel et al. "Poincaré embeddings for learning hierarchical representations" learns embedding, but if we are not mistaken, the paper does not prove that the result is a distance in the original space of HS-trees.  Contrary,  the distance proposed in our work has parameters and we guarantee that for all their values in feasible range the requirements on distance (triangular inequality) are satisfied.
>
> * "Hyperbolic graph neural networks" of Nickel et al. and later the works of Chami et al. "Hyperbolic graph convolutional neural networks" proposes to adapt graph neural networks to use hyperbolic (Poincare and Lorenthz embedding), but again it is not clear how is this related to the definition of a distance on HS-trees.
>
> * "From Trees to Continuous Embeddings and Back: Hyperbolic Hierarchical Clustering" and "Gradient-based hierarchical clustering using continuous representations of trees in hyperbolic space" both proposed a clustering method improving about linkage clustering. But they are not about distances between samples to be clustered, as is our work. We admit we could use Hyperbolic Hierarchical Clustering instead of single-linkage clustering, but single-linkage clustering was meant only for demonstration. The input to Hyperbolic Hierarchical Clustering would on the end be the same distance matrix as to single-linkage clustering, computed by our method.
>
> With respect to this, can you be little bit more specific how exactly should we compare to the algorithms you have mentioned?
>
> **Is there any hope to prove something about the quality of the metric found?**
>
>  What do you mean exactly by the quality of the metric? The quality is likely application specific. The properties of the metric are mostly influence by a distance on bags, which is discussed in the paper. If for a given problem bags are always sets and never multi-sets, one can use cheaper set-distances instead of expensive distances on multi-sets. Besides, we have found important for practice to weight contributions of sub-trees. But this weighting is similar to weighting of coordinates in Euclidean spaces, as is done in Mahalanobis  distance with diagonal weight (covariance) matrix.

---

### Meta-Review · Area_Chair_qFV6 · 2024-12-20

**Metareview:**

This paper presents a novel metric for comparing structured data in formats like JSON and XML: Hierarchically-structured Tree Distance (HTD). This metric is modular and differentiable and can be used in various tasks such as classification, clustering, and anomaly detection. In the experimental evaluation the metric behaves competitively with neural network methods with less parameters and the authors mention that the method is more suited to analyze heterogeneous Graph Neural Networks than Tree Mover’s Distance.


Strengths:
- the definition of appropriate metrics for hierarchies is an important question,
- new distance for tree-structured data that generalizes Tree Mover's distance making it modular and differentiable,
- paper well written.

Weaknesses:
- the theory appears straightforward,
- lack of consideration of related works,
- the contribution appears limited,
- lack of discussion on paper's limitations.

During rebuttal authors provided answers to 3 on the 4 reviews. The answers did not convince the reviewers to change their evaluation, all of them rated the paper from 3 to 5. There is a clear consensus for saying that the contribution is not sufficient for ICLR in its current form.

Rejection is then proposed.

**Additional Comments On Reviewer Discussion:**

The situation was very clear: all reviewers evaluated the paper below the acceptance bar. The answers provided did not convince the reviewers to change their evaluation. There is thus a consensus for rejection.

---

### Decision · Program_Chairs · 2025-01-22

Reject